# Apigenin Ameliorates Hyperuricemia and Renal Injury through Regulation of Uric Acid Metabolism and JAK2/STAT3 Signaling Pathway

**DOI:** 10.3390/ph15111442

**Published:** 2022-11-21

**Authors:** Tianyuan Liu, Huimin Gao, Yueyi Zhang, Shan Wang, Meixi Lu, Xuan Dai, Yage Liu, Hanfen Shi, Tianshu Xu, Jiyuan Yin, Sihua Gao, Lili Wang, Dongwei Zhang

**Affiliations:** 1Diabetes Research Center, Traditional Chinese Medicine School, Beijing University of Chinese Medicine, Beijing 100029, China; 2Institute of Chinese Materia Medica, China Academy of Chinese Medical Science, Beijing 100700, China; 3Department of TCM Pharmacology, Chinese Material Medica School, Beijing University of Chinese Medicine, Beijing 102488, China

**Keywords:** hyperuricemia, apigenin, uric acid, molecular docking, JAK2/STAT3 signaling, renal injury

## Abstract

Hyperuricemia (HUA) is a kind of metabolic disease with high incidence that still needs new countermeasures. Apigenin has uric-lowering and kidney-protective activities, but how apigenin attenuates HUA and renal injury remains largely unexploited. To this end, an acute HUA mouse model was established by intraperitoneal injection of potassium oxazinate and oral administration with hypoxanthine for 7 consecutive days. Apigenin intervention decreased serum uric acid (UA), creatinine (CRE), blood urea nitrogen (BUN), interleukin-1β (IL-1β), interleukin-6 (IL-6), tumor necrosis factor (TNF-α), interleukin-18 (IL-18), liver xanthine oxidase (XOD), and urine protein levels, and increased serum interleukin-10 (IL-10) and urine UA and CRE levels in HUA mice. Moreover, administration of apigenin to HUA mice prevented renal injury, decreased renal glucose transporter 9 (GLUT9) and urate anion transporter 1 (URAT1) levels, and increased renal organic anion transporter 1 (OAT1). These alterations were associated with an inhibition of IL-6, phospho-janus kinase 2 (P-JAK2), phospho-signal transducer, and activator of transcription 3 (P-STAT3), and suppression of cytokine signaling 3 (SOCS3) expression in the kidneys. Additionally, the molecular docking results showed that apigenin had strong binding capacity with UA transporters and JAK2 proteins. In summary, apigenin could improve UA metabolism and attenuate renal injury through inhibiting UA production, promoting excretion, and suppressing the JAK2/STAT3 signaling pathway in HUA mice. The results suggest that apigenin may be a suitable drug candidate for management of HUA and its associated renal injury.

## 1. Introduction

Hyperuricemia (HUA) is characterized by sustained high levels of serum uric acid (UA), which is caused by overproduction or underexcretion of UA or both in the body [1,2,3]. It is one of the main contributors to the development of gout, hypertension, cardiovascular disease, atherosclerosis, diabetes, and dyslipidemia [4,5,6,7,8,9,10], and also the four metabolic disorders that seriously affect body health and life quality [11,12]. The risk factors, including genetic factors, ages, lifestyles, obesity, and drug history, contribute to the prevalence of HUA [13,14,15]. An epidemiological study has shown that the incidence of HUA among Chinese adults during 2018–2019 was about 14.0%, which was increased by 2.9% compared with 2015–2016 [16]. In addition, the populations of HUA sufferers recently tend to be younger in the world [17]. However, the current countermeasures, including dietary intervention and medications, may not meet the needs of the patients and society owing to low compliance and high risk of adverse events [18,19]. There is emerging evidence indicating that natural products may benefit for UA metabolism with lower risk of side effects, and shed light on seeking anti-HUA drug candidates [20,21].

Serum UA levels are mainly determined by: (1) liver xanthine oxidase (XOD), which is a rate-limiting enzyme that converts hypoxanthine to xanthine and then xanthine to UA [22]; (2) renal urate anion transporter 1 (URAT1), glucose transporter 9 (GLUT9), and organic anion transporter 1 (OAT1), which are responsible for renal UA reabsorption and excretion [23]. The over-accumulation of serum UA caused by its overproduction and/or underexcretion may aggravate inflammation response [24,25,26,27] and subsequently disturb renal function [28]. Moreover, the pro-inflammatory cytokines, such as interleukin-6 (IL-6), interleukin-1β (IL-1β), tumor necrosis factor (TNF-α), and interleukin-18 (IL-18), contribute to boosting inflammatory response and HUA nephropathy through activation of janus kinase 2 (JAK2)/signal transducer and activator of transcription 3 (STAT3) signaling pathway [29,30,31]. In addition, as a downstream regulator of JAK/STAT signaling pathway, suppressor of cytokine signaling 3 (SOCS3) may aggravate renal injury in response to UA overproduction [32,33]. Therefore, alleviation of inflammation response may offer benefits for HUA management and renal health.

Apigenin (4,5, 7-trihydroxyflavone), a natural flavonoid, is mainly derived from Apium graveolens L.(celery) and it is also found in a variety of plants, fruits, and vegetables [34]. The accumulating evidence suggests that apigenin may alleviate UA in purine bodies or potassium oxonate (PO)-induced HUA mice [35,36]. In addition, apigenin may attenuate HUA-induced renal fibrosis via inhibition of the Wnt/β-catenin pathway [11]. Interestingly, apigenin was demonstrated to exert protective effect on renal ischemia/reperfusion injury (IRI) in rats through regulation of the JAK2/STAT3 signaling pathway [37]. This prompts us to look into whether and how apigenin regulates JAK2/STAT3 signaling in the development of HUA and its associated renal injury. For this purpose, in the present study, the PO and hypoxanthine (Hypox)-induced acute HUA mouse model was established, and the action of apigenin on serum UA metabolism and renal JAK2/STAT3 signaling pathway were examined. This study will provide new insights into the mechanisms underlying the pharmacological effects of apigenin on acute HUA and highlight the potentials of apigenin in management of patients with hyperuricemic nephropathy.

## 2. Results

### 2.1. Apigenin Reduces Serum Levels of UA, CRE, and BUN, and Increases Urine Levels of UA and CRE in HUA Mice

As shown in Figure 1A–C, serum levels of UA, creatinine (CRE), and blood urea nitrogen (BUN) in the HUA model control (HC) group were significantly increased compared to those in the normal control (NC) group (*p* < 0.01). As expected, treatment with apigenin (25, 50 and 100 mg/kg) markedly decreased serum levels of UA and BUN in HUA mice in comparison with the vehicle-treated ones (*p* < 0.01). However, only a high dose of apigenin (100 mg/kg) could decrease serum CRE levels in HUA mice. In addition, serum UA levels were markedly decreased in allopurinol (Allop)-treated HUA mice in comparison with the vehicle-treated ones (*p* < 0.01), but this treatment did not affect serum BUN and CRE levels.

As shown in Figure 1D–I, on day 0, there were no significant differences in the urine levels of UA and CRE between each group. However, 2 or 6 days after exposure to PO and Hypox, the urine UA levels of the mice in the HC group showed a decreasing trend in comparison with those in the NC group. In addition, the urine CRE levels in the HC group were significantly decreased compared to those in the NC group. As expected, oral administration with apigenin (25, 50, and 100 mg/kg) for 2 or 6 days markedly increased urine levels of UA and CRE in HUA mice compared to those in the vehicle-treated ones (*p* < 0.05 or 0.01). However, Allop intervention only significantly increased urine CRE levels on day 6 in HUA mice (*p* < 0.05). These results suggest that apigenin not only has dual actions on UA metabolism, but also could alleviate HUA-induced renal injury.

### 2.2. Apigenin Ameliorates Renal Pathological Microstructures and Urine Protein Levels in HUA Mouse

As shown in Figure 2A, the kidney appears more grey-redish color and touches harder in the mice of the HC group than in the NC group. As expected, treatment with apigenin or Allop almost reversed the color alterations and the texture in the kidneys in HUA mice. As shown in Figure 2B, the relative weight of kidney (kidney weight (KW)/body weight (BW)) in the HC group was higher than those in the NC group (*p* < 0.01). Administration with apigenin could prevent an increase in the KW/BW in HUA mice compared to the vehicle-treated ones (*p* < 0.05 or 0.01). In addition, as shown in Figure 2C, Hematoxylin/eosin (H&E) staining showed that the kidney showed vacuolization of glomerular epithelium, edema and interstitial hemorrhage, tubular dilation, and infiltration of inflammatory cells in the mice of the HC group. Moreover, as shown in Figure 2D,E, the urine albumin levels and their relative value to CRE of mice in the HC group were significantly increased compared with those in the NC group (*p* < 0.01). Furthermore, as shown in Figure 2F,G, the Masson trichome staining showed that collagen fibers were over-accumulated in the tubule interstitium and glomeruli in the mice of the HC group (*p* < 0.01). As expected, treatment with apigenin or Allop significantly reversed these alterations in HUA mice (*p* < 0.05 or 0.01). These results suggested that apigenin could alleviate renal injury in HUA mice.

### 2.3. HUA Exposure and Apigenin Intervention Do Not Affect Liver Function and Pathological Microstructures in Mice

As shown in Figure 3A–D, no significant changes were observed in the relative liver weight to body weight (LW/BW) among the different groups of mice. H&E staining also showed that no obvious alterations in the livers of each group of mice. In addition, there were no significant differences in serum levels of aspartate aminotransferase (AST) and alanine aminotransferase (ALT) among different groups of mice (Figure 3B,C).

### 2.4. Apigenin Suppresses Serum Inflammatory Response in HUA Mice

Accumulating evidence suggests that acute HUA may cause inflammation response, which further upregulates serum UA levels and subsequently causes kidney dysfunction [38,39,40]. As shown in Figure 4A–D, serum levels of IL-6, IL-1β, TNF-α, and IL-18 in the HC group of mice were significantly increased compared to those in the NC group (*p* < 0.01). Notably, administration with apigenin (25, 50, and 100 mg/kg) and Allop to HUA mice significantly reduced serum levels of IL-6, TNF-α, and IL-18 in comparison with those of the vehicle-treated ones (*p* < 0.01). However, as shown in Figure 4B, only the medium and high doses of apigenin treatment significantly decreased serum IL-1β levels in HUA mice in comparison with those of the vehicle-treated ones (*p* < 0.05 or 0.01). In addition, as shown in Figure 4E, serum interleukin-10 (IL-10) levels in the mice of the HC group were markedly decreased compared to those in the NC group (*p* < 0.01). Notably, 100 mg/kg of apigenin significantly prevented a decrease in serum IL-10 levels in HUA mice (*p* < 0.05).

### 2.5. Apigenin Decreases Liver XOD Levels and Renal Expression Levels of GLUT9 and URAT1, and Increases Renal Expression Levels of OAT1 in HUA Mouse

Next, we examined the effect of apigenin on the protein expressions involved in UA generation and excretion in HUA mouse. As shown in Figure 5A, the liver XOD levels in the HC group were higher than those in the NC group. Notably, administration with apigenin (100 mg/kg) or Allop to HUA mice significantly decreased liver XOD levels in comparison with those with the vehicle-treated ones (*p* < 0.05 or 0.01). Based on the above-mentioned results, the high dose of apigenin (100 mg/kg) was better than the low and medium doses of apigenin in preventing the alterations in HUA mice. Therefore, the mice in the high dose of apigenin-treated group were used to the following experiments.

Moreover, as shown in Figure 5B–D, western blot results revealed that the expression levels of GLUT9 and URAT1 were significantly increased, and the expression levels of OAT1 were markedly decreased in the kidneys of the HC group compared to those in the NC group (*p* < 0.05). As expected, apigenin (100 mg/kg) markedly reversed the expression levels of GLUT9, URAT1, and OAT1 in the kidneys of HUA mice (*p* < 0.05). These results suggested that apigenin may have the ability of promoting UA excretion in the kidney.

### 2.6. Interactions between Apigenin Molecule with UA Transporters and JAK2 Protein by Molecular Docking

In order to examine the binding ability of apigenin with UA transporters and JAK2 protein, the SYBYL-X 2.0 software was used to conduct molecular docking analysis. As shown in Figure 6G–J and Table 1, the results revealed that apigenin and XOD protein structure had 5 binding sites (the active site residues were THR354, ASN351, VAL259, GLY260, and SER347). The results of Polar score (3.89 points), absolute value of Grash score (−1.24 points), and C-score (3 points) indicate that apigenin and XOD protein could produce high energy during the collision. The total score (6.20 points) also shows that apigenin molecule has a strong binding capacity with XOD protein.

There were two binding sites between apigenin with GLUT9 (the active site residues were LYS431 and GLY58) and URAT1 (the active site residues were LYS393 and HIS245) proteins, respectively. Though the total score (GLUT9: 3.87 points; URAT1: 4.90 points) was not that high, their C-score was 5 points, and they had relatively low absolute Grash scores (GLUT9: −0.76 points; URAT1: −0.34 points) and Polar scores (GLUT9: 1.16 points; URAT1: 2.49 points), indicating that apigenin has a strong binding capacity with the two proteins.

There were 3 binding sites (the active site residues were ASN35, TYR141, and SER203) between apigenin and OAT1 protein, and the total score was over 6 points, while the C-score was 3 points. Additionally, the smaller Polar score value (2.61 points) and the absolute value of Grash score (−0.65 points) indicate that the molecule was closer to the active center of the target OAT1 protein and had the higher energy generated by collision. Therefore, apigenin has a strong binding capacity with OAT1 protein.

In addition, as shown in Figure 6K, the docking analysis results revealed that apigenin molecule had 3 binding sites (the active site residues were GLU930, LEU932, and GLU898) with JAK2 protein, and the total score (6.48 points), C-score (5 points), Grash score (−1.38 points), and Polar score (3.50 points) indicated that apigenin has a strong binding capacity with JAK2 protein. Together, the molecular docking results suggested that apigenin may act on UA transporters and JAK2 protein.

### 2.7. Apigenin Inhibits JAK2/STAT3 Signaling Pathway in HUA Mouse

As shown in Figure 7A–F, the results from the immunohistochemistry (IHC) staining revealed that the expression levels of IL-6, phospho-janus kinase 2 (P-JAK2), and phospho-signal transducer and activator of transcription 3 (P-STAT3) were significantly elevated in the kidneys in the HC group of mice compared to those in the NC group (*p* < 0.01). By contrast, supplement of 100 mg/kg of apigenin to HUA mice significantly down-regulated the expression levels of IL-6, P-JAK2, and P-STAT3 compared to those of the vehicle-treated ones (*p* < 0.05 or 0.01). Moreover, the results from the western blots corroborated the findings from the IHC staining.

The activation of JAK2/STAT3 may facilitate SOCS3 expression, which potentiates the inflammatory response in the kidney [32,45]. Therefore, as shown in Figure 7G–J, the SOCS3 expression levels were increased in the kidneys of the HC group. As expected, 100 mg/kg of apigenin treatment significantly reversed renal SOCS3 expression levels in HUA mice compared to those of the vehicle-treated ones (*p* < 0.05). These results suggested that apigenin may inhibit JAK2/STAT3 signaling pathway in the kidney of HUA mouse.

## 3. Discussion

The globally increasing prevalence of HUA and its associated nephropathy are in high demand of new countermeasures [46]. Here, we have shown that apigenin could decrease serum UA, CRE, and BUN levels, increase urine CRE levels, and decrease urine protein levels as well as improve renal pathological microstructures in this acute HUA mouse model. We also found that administration of HUA mice with apigenin preserves serum levels of IL-6, IL-1β, TNF-α, IL-18, and IL-10. In addition, we have provided the evidence of the actions of apigenin on HUA mice, including a decrease in the liver XOD and renal GLUT9 and URAT1 expressions, and an increase in OAT1 expression, as well as an inhibition of the JAK2/STAT3 signaling pathway (Figure 8).

In the current study, apigenin was demonstrated to decrease serum UA, CRE, and BUN levels, and increase urine UA and CRE levels in PO- and Hypox-induced acute HUA mouse. In support of the current findings, apigenin was reported to reduce serum levels of UA, CRE, and BUN in PO- and adenine-induced hyperuricemic nephropathy KM mice [11], and decrease UA production in AML12 hepatocytes [36]. As is well known, inappropriate high levels of serum UA may result in increased generation, or decreased excretion, or a combination of both. Interestingly, we have found that apigenin could reduce liver XOD activity in HUA mice. Moreover, apigenin was able to promote the excretion of UA and CRE in urine through decreasing apical URAT1, basolateral GLUT9 expressions, and increasing OAT1 expression. The docking analysis results also revealed that apigenin has a strong affinity with the proteins involved in UA metabolism, including XOD, GLUT9, URAT1, and OAT1. Collectively, these results suggested that apigenin may not only inhibit UA production but also promote UA excretion, and thus have a potential to be developed as one kind of anti-HUA drug candidates.

The sustained high serum UA levels may cause renal endothelial cell inflammation, and an enhanced secretion of pro-inflammatory molecules such as IL-6, TNF-α, and IL-1β and a diminished production of anti-inflammatory molecules, such as IL-10 [47], as well as an increase in renal injury, which have been demonstrated in this acute HUA mouse model. Interestingly, apigenin intervention inhibited HUA-induced inflammation and renal injury exhibited by regulating CRE, BUN, and urine protein levels, and inhibiting collagen overproduction via regulation of IL-6, TNF-α, IL-1β, and IL-10 secretion. Similarly, apigenin has been demonstrated to reduce IL-6, TNF-α, and IL-1β levels, and increase IL-10 levels in the kidneys or serum in doxorubicin-induced male nephropathy BALB/c mice [48] or in furan-induced nephropathy mice [49]. In addition, apigenin-solid lipid nanoparticles (SLNPs) to reduce renal mRNA expression levels of IL-6, TNF-α, and IL-1β in streptozotocin-induced diabetic nephropathy rats [50]. Together, our current findings in conjunction with the above-mentioned investigations from other groups indicated that apigenin could protect renal injury through inhibition of inflammation.

We have for the first time to report that apigenin intervention could decrease SOSC3 expression in the kidney of HUA mouse. The accumulating evidence suggest that SOSC3 is a downstream regulator of JAK2/STAT3 signaling pathway [51,52], which is involved in inflammation response. Indeed, we have found that this alteration is positively associated with downregulation of JAK2 and STAT3 phosphorylation in the kidneys of HUA mice in response to apigenin treatment. In addition, we have found that apigenin could downregulate renal IL-6 expression in this acute HUA mouse model. Moreover, it is demonstrated that IL-6 may potentiate inflammatory response through activation of the JAK2/STAT3 signaling pathway [53]. In parallel with this line of evidence, apigenin was reported to decrease renal IL-6 expression in diabetes [50] or doxorubicin-induced nephropathy [48]. In addition, apigenin was able to reverse drug resistance through suppression of STAT3 signaling in MCF-7 and MCF-7/ADR cells [54]. The docking analysis results also revealed that apigenin have a high affinity with JAK2, and may form hydrogen bonds with Leu932, Glu930, and Glu898 [55], thus inhibiting JAK2 activity. Therefore, it is reasonable to conclude that apigenin may ameliorate HUA and renal injury through regulation of the JAK2/STAT3 signaling pathway.

In the present study, we demonstrated that apigenin does not affect liver functions in acute HUA mice. However, Singh et al. reported that acute exposure of apigenin may cause liver dysfunction in mice [56]. The inconsistent findings between these two studies may be explained as follows: (1) The species and ages of experimental mice are different. In the present study, 8-week-old (18–22 g) male Kunming (KM) mice were employed to examine liver ALT and AST levels and histo-microstructures. However, in the previous investigation conducted by Singh et al., 10–12-week-old (25–30 g) male Swiss mice were used to observe the liver toxicity; (2) the different administration routes may influence the metabolism and bioavailability of apigenin. In the present study, apigenin was orally gavaged to the HUA mouse, while in the previous investigation, apigenin was administered to the mouse by intraperitoneal injection.

In the present study, the urine levels of UA were determined on days 0, 2, and 6 during modeling and drug administration, which may allow the efficacy of apigenin on the alterations of urine UA and CRE in mice to be better observed [57,58,59,60] In addition, the present investigation aims to study the actions of apigenin on acute HUA-induced by PO and Hypox. Therefore, apigenin was simultaneously administrated after each modeling to exclude possible experimental errors caused by animal self-recovery. The results revealed that apigenin may prevent the development of acute HUA in mice. Interestingly, Li et al. [11] reported that apigenin may alleviate chronic HUA and its associated nephropathy in mice induced by PO and adenine. Taken together, these findings suggested that apigenin may be used in the management of acute and chronic HUA in clinical trials.

The present results showed that the urine levels of UA and CRE were not consistent before and after the treatment in each group. These alterations may be attributed to the following two reasons: (1) The vehicle-treatment effect. During the experiments, mice in the NC group were intraperitoneally injected and/or gavaged with the corresponding control vehicles. These treatments may induce a certain type of psychological and/or physical stresses, which lead to the variations in the values of blood and urine parameters. Therefore, we have set the controls for each time point to eliminate possible errors between different batches. Indeed, several investigations have shown that blood UA levels may increase under acute stress [61,62,63], which results in a decrease in UA or CRE excretion; (2) we did not collect the whole volume of daily urine. In the present study, we have only harvested the certain interval of urine from the mouse.

In this study, Allop was demonstrated to decrease the levels of liver XOD and serum UA, and increase the levels of urine CRE (day 6) in this rapid HUA mouse model. However, Allop did not reverse the increased serum levels of CRE and BUN in HUA mice. By contrast, Chen et al. [64] reported that Allop was able to decrease serum levels of CRE and BUN in HUA mice. The underlying cause to explain the conflicting findings may be attributed to the differences in the approaches to establish HUA mouse model. In their study, the authors employed intraperitoneal injection with PO to establish the HUA mouse model. However, in the present study, Hypox (150 mg/kg, gavage) was used to induce HUA mice in addition to PO exposure.

Some limitations of the present study may exist regarding interpretation of the data. Firstly, we did not evaluate the anti-inflammation effect of apigenin on renal tubular epithelial cells. However, apigenin has been demonstrated to inhibit the overproduction of TNFα, IL-1β, and IL-6 in D-glucose treated HK-2 cells [65]. Secondly, we did not employ a JAK2 inhibitor to examine the direct interactions between apigenin and JAK2. Nevertheless, the docking analysis, IHC staining, and western blot results demonstrated that apigenin could inhibit JAK2 phosphorylation. In addition to that, apigenin was demonstrated to suppress JAK2 and STA3 phosphorylation in breast cancer and hepatocellular carcinoma cells [66,67]. Despite these concerns, this study may indicate that apigenin may alleviate HUA and renal injury through the regulation of the JAK2/STAT3 signaling pathway. However, the underlying mechanism behind the anti-HUA effect of apigenin still merits further investigations.

## 4. Materials and Methods

### 4.1. Reagents and Antibodies

Apigenin (purity ≥ 98%, CAS:520-36-5) was purchased from RuiFenSi Biotechnology Co., Ltd. (Chengdu, China). Allop (H31020334) was bought from Xinyi Mixane Pharmaceutical Co., Ltd. (Shanghai, China). PO (CAS:2207-75-2) and Hypox (CAS:68-94-0) were purchased from Macklin (Shanghai, China). Antibodies against JAK2 (Cat# 3023), STAT3 (Cat# 30835), and P-STAT3 (Cat# 9145) were obtained from Cell Signaling (Danvers, MA, USA). Antibodies against URAT1 (Cat# 14937-1-AP), OAT1 (Cat# 26574-1-AP), GLUT9 (Cat# 26486-1-AP), IL-6 (Cat# 21865-1-AP), SOCS3 (Cat# 66797-1-Ig), β-actin (Cat# 66009-1-Ig), and GAPDH (Cat# 60004-1-Ig) were obtained from Proteintech Co., Ltd. (Rosemont, IL, USA). Antibodies against P-JAK2 (Cat# AP0531) were obtained from ABclonal company (Wuhan, China). The commercial kits for UA (Cat# C012-2-1), CRE (cat# C011-2-1), BUN (Cat# C013-2-1), urinary albumin (C035-2-1), ALT (Cat# C009-2-1), AST (Cat# C010-2-1), and XOD (Cat# A002-1-1) were purchased from Nanjing Jiancheng Bioengineering Institute (Nanjing, China). The ELISA kits for IL-6 (KT2163-A), IL-10 (KT2176-A), IL-18 (KT2169-A), IL-1β (KT2040-A), and TNF-α (KT2132-A) were purchased from Jiangsu KeTe Biotechnology Co., Ltd. (Nanjing, China).

### 4.2. Animals and Treatments

Male KM mice (20 ± 2 g) were purchased from Beijing SiBeiFu Animal Technology Co., Ltd. (Beijing, China) and housed in a specific-pathogen-free (SPF) animal facility with constant temperature (21 ± 2 °C), humidity (55 ± 5%), and a 12-h light/dark cycle at the Beijing University of Chinese Medicine (BUCM). All mice were supplied with regular chow and drinking water ad libitum. All the animal procedures were approved by the BUCM Animal Care Committee, Beijing, China (Approve No. BUCM-4-2021042302-2016).

After acclimatization for one week, the KM mice were randomly divided into six groups (10 mice per group): (1) NC group, (2) HC group, (3) Allop group, (4) low dose of apigenin-treated (APL) group, (5) medium dose of apigenin-treated (APM) group, (6) high dose of apigenin-treated (APH) group. In detail, mice in the last five groups were daily administered with PO (300 mg/kg, intraperitoneal injection) and Hypox (150 mg/kg, gavage) according to the previous publications [57,58,68]. Subsequently, the mice in the Allop group were orally administrated with allopurinol (5 mg/kg). The mice in the APL, APM and APH groups were orally administrated with 25, 50, and 100 mg/kg of apigenin, respectively. The mice in the NC and HC groups were administrated with the same volume of the vehicles. One hour after the induction of PO and Hypox, drugs or corresponding vehicles were orally administered for 7 days. The experimental protocol is illustrated in Figure 9.

During the treatment, the urine of the mouse on days 0, 2, and 6 were collected for subsequent analysis. One hour after the last treatment, blood was collected from the anesthetized mouse to obtain the serum. The liver and kidney were quickly removed from the mouse and then stored either in −80 °C or in 4% paraformaldehyde solution.

### 4.3. Biochemical Analysis

The levels of UA, CRE, BUN, albumin, ALT, and AST in serum or urine, and XOD in the liver were measured using commercial kits according to the manufacturer’s instructions. Serum levels of IL-6, IL-1β, IL-18, TNF-α, and IL-10 were detected using ELISA kits according to the corresponding protocols.

### 4.4. H&E and Masson Trichome Staining

The kidneys from the mice were fixed in 4% paraformaldehyde for 48 h and then embedded in paraffin. The H&E and Masson trichome staining was performed according to the routine procedure [11,69]. The pathological sections were imaged with an Olympus BX53 Fluorescence microscope (Tokyo, Japan).

### 4.5. IHC Staining

The IHC staining of the kidney was conducted as previously described [70]. Briefly, the slides were incubated with the appropriate primary antibody IL-6 (1:250), P-JAK2 (1:100), and P-STAT3 (1:200) overnight at 4 °C, followed by the corresponding secondary antibodies and DAB staining solutions. Subsequently, slides were examined and photographed using an Olympus BX53 microscopy. The intensity of DAB staining was analyzed using Image Pro Plus 6.0 software and expressed as an integrated optical density (IOD) value.

### 4.6. Western Blot Assay

The total proteins of the kidneys were extracted with a RIPA buffer and quantified by a BCA kit. The equal amounts of protein were loaded onto SDS-PAGE gels and subsequently transferred to the PVDF membrane. Then, the membrane was separately incubated with the appropriate primary antibody [IL-6 (1:500), JAK2 (1:1000), P-JAK2 (1:500), STAT3 (1:1000), P-STAT3 (1:2000), OAT1 (1:1000), URAT1 (1:500), GLUT9 (1:1000), and SOCS3 (1:2000)] overnight at 4 °C and subsequently incubated with the corresponding secondary antibodies for 1 h at room temperature. Immunopositive bands were visualized using high sensitivity ECL and images were captured by the Azure (C500) Bio-image system (Azure Biosystems Inc, Dublin, USA). The grey values of the bands were quantified using an Image J software (V1.8.0.112) and normalized with the corresponding β-actin (1:10,000) or GAPDH (1:10,000).

### 4.7. Molecular Docking

The 3D structure of apigenin molecule was retrieved from the PubChem (https://pubchem.ncbi.nlm.nih.gov/, accessed on 10 May 2022). The crystal structures of the XOD (PDB code, 1FIQ), GLUT9 (PDB code, 4GBY), URAT1 (PDB code, AF-Q96S37-F1), OAT1 (PDB code, AF-Q4U2R8-F1), and JAK2 (PDB code, 3UGC) were derived from the RCSB Protein Data Bank (https://www.rcsb.org/, accessed on 10 May 2022), National Center for Biotechnology Information (https://www.ncbi.nlm.nih.gov/, accessed on 10 May 2022) and UniProt (https://www.uniprot.org/, accessed on 10 May 2022). The Sybyl-x 2.0 software was used for molecular docking analysis of apigenin with UA transporters and JAK2 proteins.

### 4.8. Statistical Analysis

A GraphPad Prism was used to analyze the original data. All the data were presented as mean ± SEM. When the data met homogeneity of variance and normality, a one-way analysis of variance (ANOVA) was employed. A Dunnett’s T3 test or nonparametric test was applied, respectively, when the date met a normal distribution, but the homogeneity of variances was not achieved, or did not meet a normal distribution. A value of *p* < 0.05 was considered to be a statistical difference.

## 5. Conclusions

In summary, the current study provides evidence that apigenin could attenuate UA levels and renal injury in HUA mice. The potential mechanism behind this action may be related to an inhibition of UA overproduction and a promotion of UA excretion as well as the regulation of JAK2/STAT3 signaling pathway. These results highlight the potential of apigenin in the management of HUA patients with renal injury. Further investigations of the actions of apigenin HUA will contribute to providing a suitable scaffold for developing multitargeted anti-HUA agents.

## Figures and Tables

**Figure 1 pharmaceuticals-15-01442-f001:**
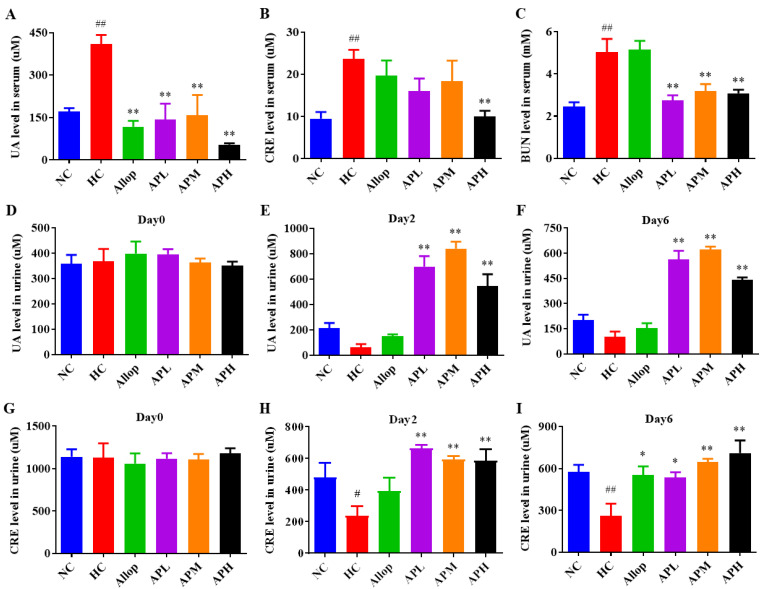
Apigenin reduces serum levels of UA, CRE, and BUN, and increases urine levels of UA and CRE in HUA mice. Serum levels of UA (**A**; μM, *n* = 10), CRE (**B**; μM, *n* = 10), and BUN (**C**; mM, *n* = 10). Urine levels of UA on days 0, 2, and 6 (**D**–**F**; μM, *n* = 10, 6 and 10). Urine levels of CRE on days 0, 2, and 6 (**G**–**I**; μM, *n* = 10, 10 and 7). Allop denotes allopurinol-treated group, APL denotes low dose of apigenin-treated group, APM denotes medium dose of apigenin-treated group, APH denotes high dose of apigenin-treated group. ^##^
*p* < 0.01, ^#^
*p* < 0.05 vs. the NC group. ** *p* < 0.01, * *p* < 0.05 vs. the HC group.

**Figure 2 pharmaceuticals-15-01442-f002:**
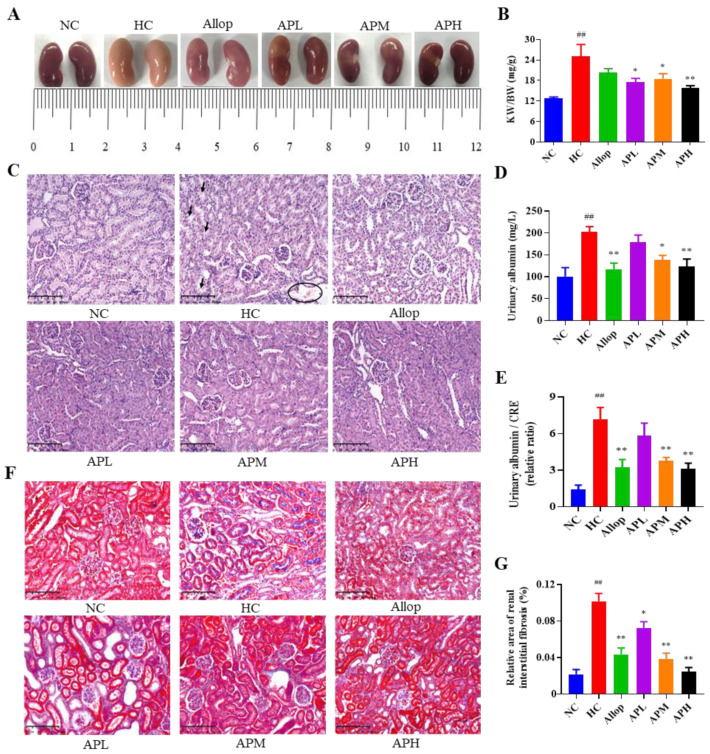
Apigenin ameliorates renal general appearance and pathological microstructures and urine protein levels in HUA mice. The histological appearance of the kidney (**A**). Kidney weight/body weight (**B**; KW/BW, mg/g, *n* = 10). Representative images of H&E staining (**C**; ×200, Scale bar: 100 μm) of the kidneys. Urine albumin levels (**D**; mg/L, *n* = 8). The relative levels of urinary albumin to CRE (**E**; *n* = 6). The representative images of Masson trichome staining and their analyses (**F**,**G**; ×200, Scale bar: 100 μm, *n* = 6). The black arrow indicates that the renal tubular epithelial cells were hollowed out, edema. The black circle indicates renal interstitial hemorrhage. The blue color denotes collagen fibers in the kidney. ^##^
*p* < 0.01 vs. the NC group. ** *p* < 0.01, * *p* < 0.05 vs. the HC group.

**Figure 3 pharmaceuticals-15-01442-f003:**
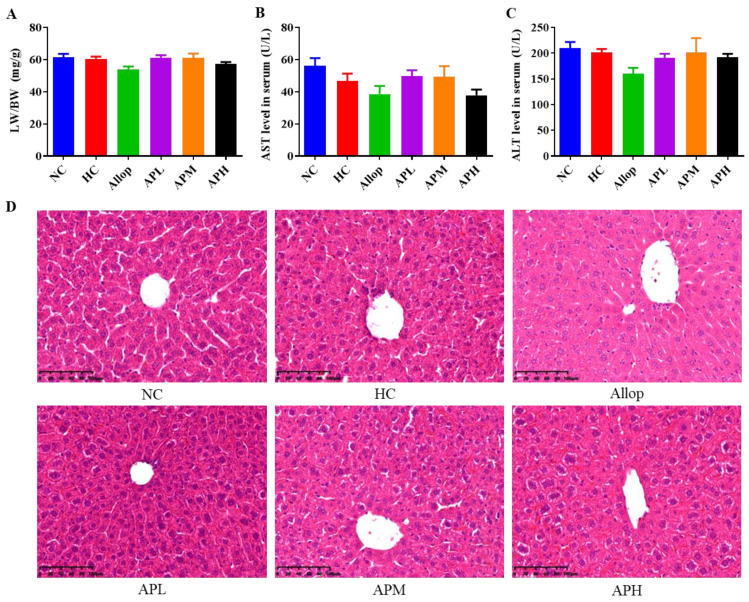
Apigenin has no effects on relative liver weight (**A**; mg/g, *n* = 10), serum levels of AST (**B**; U/L, *n* = 10) and ALT (**C**; U/L, *n* = 10), and pathological microstructures in HUA mice. Representative images of liver H&E staining (**D**; ×200, Scale bar: 100 μm).

**Figure 4 pharmaceuticals-15-01442-f004:**
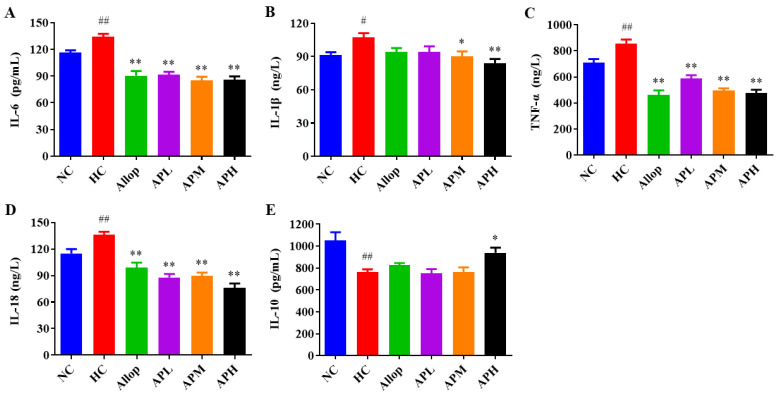
Apigenin decreases serum levels of IL-6 (**A**; pg/mL, *n* = 10), IL-1β (**B**; ng/L, *n* = 10), TNF-α (**C**; ng/L, *n* = 10), and IL-18 (**D**; ng/L, *n* = 10), and increases serum levels of IL-10 (**E**; pg/mL, *n* = 8). ^##^
*p* < 0.01, ^#^
*p* < 0.05 vs. the NC group. ** *p* < 0.01, * *p* < 0.05 vs. the HC group.

**Figure 5 pharmaceuticals-15-01442-f005:**
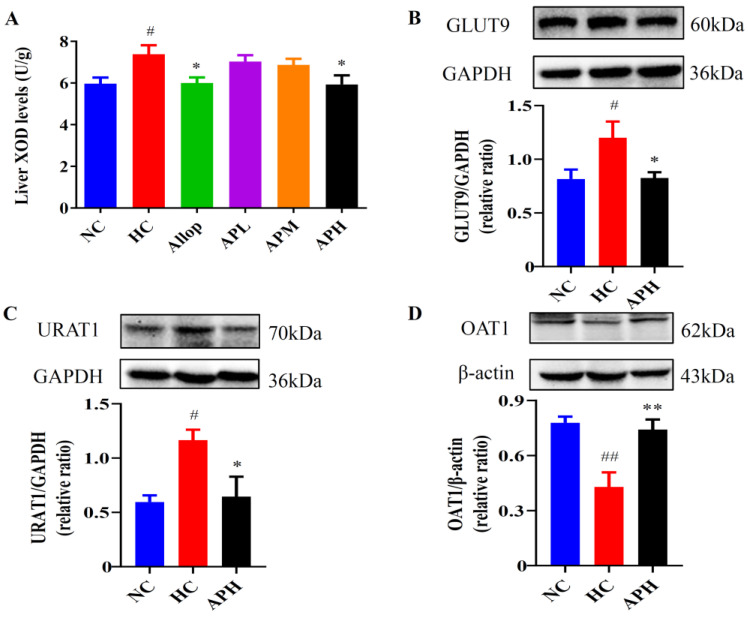
Apigenin decreases liver XOD levels (**A**; U/g, *n* =10) and renal expression levels of GLUT9 and URAT1, and increases renal expression levels of OAT1 in HUA mice. Representative images of western blot and their analyses show GLUT9 (**B**), URAT1 (**C**), and OAT1 (**D**) expressions in the kidneys (*n* ≥ 3). ^##^
*p* < 0.01, ^#^
*p* < 0.05 vs. the NC group. ** *p* < 0.01, * *p* < 0.05 vs. the HC group.

**Figure 6 pharmaceuticals-15-01442-f006:**
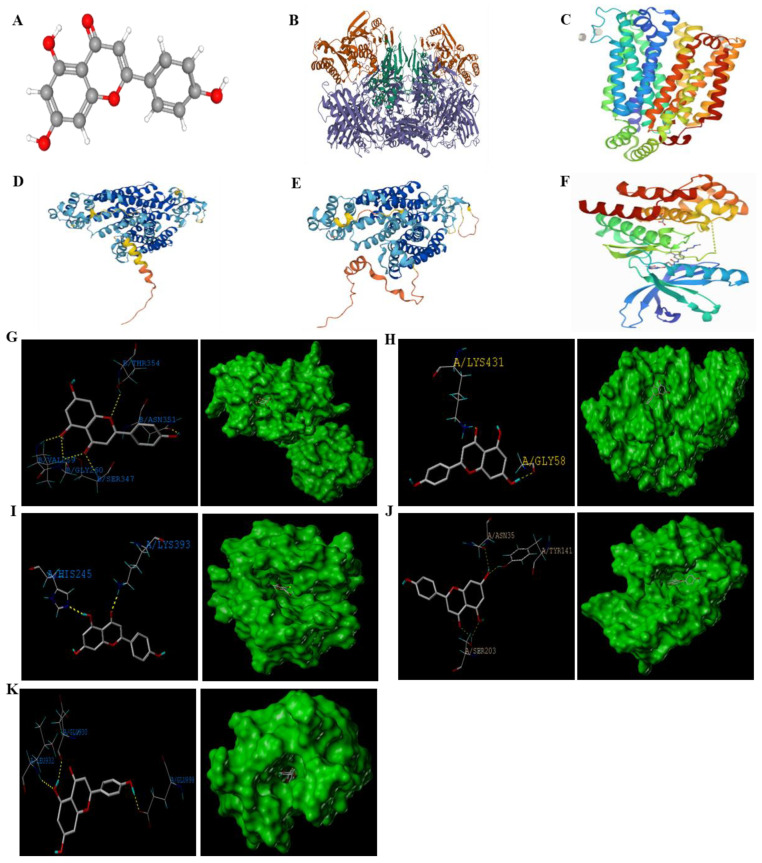
Interactions between apigenin with UA transporters and JAK2 protein by molecular docking. The 3D structure diagram of apigenin molecule (**A**). Crystal structure of XOD (PDB code, 1FIQ), GLUT9 (PDB code, 4GBY), URAT1 (PDB code, AF-Q96S37-F1), OAT1 (PDB code, AF-Q4U2R8-F1), and JAK2 (PDB code, 3UGC) proteins (**B**–**F**). Molecular docking diagram of apigenin with UA transporters and JAK2 proteins (**G**–**K**).

**Figure 7 pharmaceuticals-15-01442-f007:**
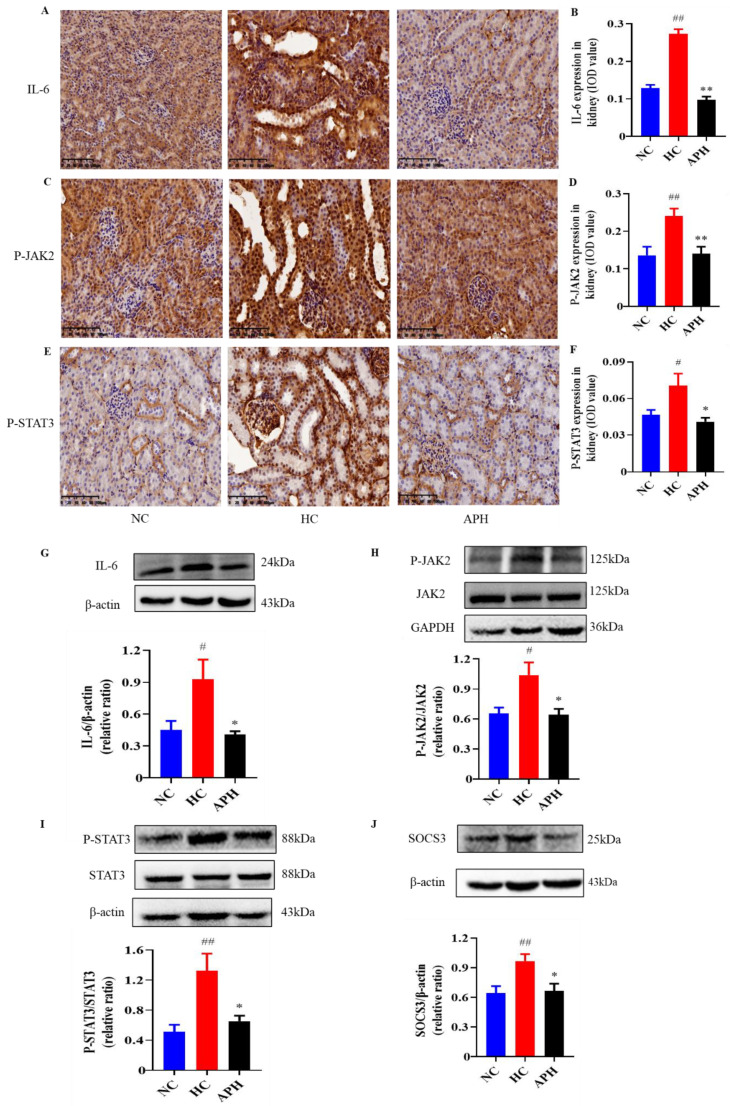
Apigenin reduces the expression levels of IL-6, P-JAK2, P-STAT3, and SOCS3 in the kidneys of HUA mice. Representative images of IHC staining (×200, Scale bar: 100 μm) and their analyses show IL-6 (**A**,**B**), P-JAK2 (**C**,**D**), and P-STAT3 (**E**,**F**) expressions in the kidneys (*n* = 5). Representative images of western blots and their analyses show IL-6 (**G**), P-JAK2 (**H**), P-STAT3 (**I**), and SOSC3 (**J**) expressions in the kidneys (*n* ≥ 3). ^##^
*p* < 0.01, ^#^
*p* < 0.05 vs. the NC group. ** *p* < 0.01, * *p* < 0.05 vs. the HC group.

**Figure 8 pharmaceuticals-15-01442-f008:**
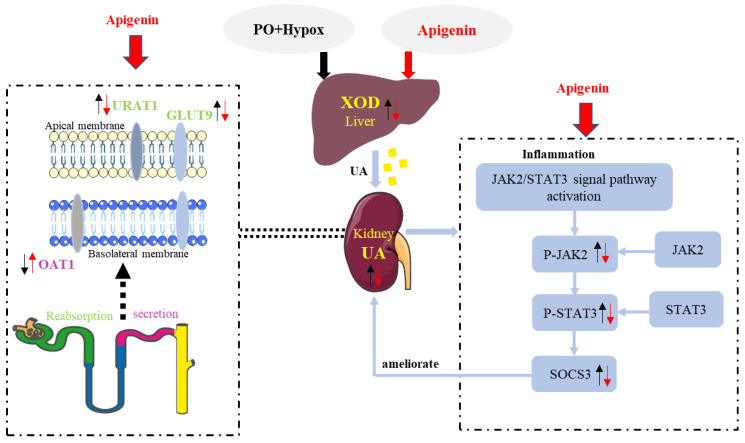
Apigenin may ameliorate HUA and renal injury through regulation of UA metabolism and the JAK2/STAT3 signaling pathway. The black arrow indicates the changes of biomarkers in HUA mice induced by PO and Hypox, and the red arrow indicates the changes of biomarkers in HUA mice after apigenin intervention.

**Figure 9 pharmaceuticals-15-01442-f009:**
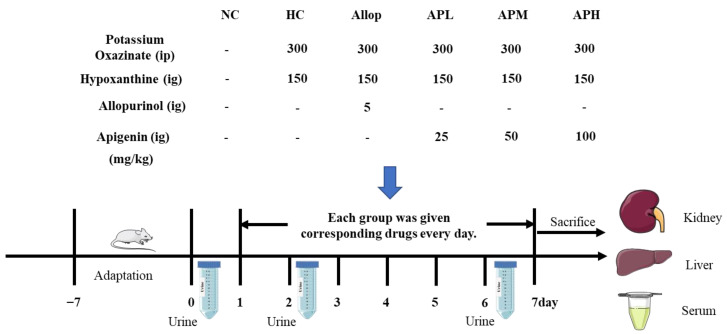
Experimental design on action of apigenin on acute HUA mice. ig denotes intragastrical administration, ip denotes intraperitoneal administration.

**Table 1 pharmaceuticals-15-01442-t001:** Interactions between apigenin with UA transporters and JAK2 protein by molecular docking.

Protein	XOD	GLUT9	URAT1	OAT1	JAK2
Crystal structure	1FIQ	4GBY	AF-Q96S37-F1	AF-Q4U2R8-F1	3UGC
Total score	6.20	3.87	4.90	6.14	6.48
Grash score	−1.24	−0.76	−0.34	−0.65	−1.38
Polar score	3.89	1.16	2.49	2.61	3.50
C-score score	3	5	5	3	5

Notes: The total score > 4.0 indicates that the molecule has a certain binding activity with the target. The total score of 5–7 indicates that the molecule and the target have a good reactivity, and the score above 7 indicates a micromole level of reactivity. The crash score reveals the inappropriate penetration into the binding site. Negative number indicates penetration. The lower the absolute value, the higher the energy. The polar score identifies the region of the ligand. The closer the molecule is to the center of the active site of the target, the smaller its value and the greater its polarity. Consensus score (C-score), is mainly used to rank the affinity of ligands bound to the active site of a receptor; the higher the value of C-score obtained by the combination of protein and molecule, the higher the reactivity [41,42,43,44].

## Data Availability

Data is contained within the article.

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
