# Peer review of "Apigenin Ameliorates Hyperuricemia and Renal Injury through Regulation of Uric Acid Metabolism and JAK2/STAT3 Signaling Pathway"

_pharmaceuticals, 2022, doi:10.3390/ph15111442_

Round 1

Reviewer 1 Report (Previous Reviewer 1)

 I would like to thank authors for addressing  my comments and suggestions,

the manuscript has been significantly improved

Reviewer 2 Report (Previous Reviewer 2)

No any further concern.

This manuscript is a resubmission of an earlier submission. The following is a list of the peer review reports and author responses from that submission.

Round 1

Reviewer 1 Report

The manuscript submitted by Liu et, al. aimed to assess the potential effect of Apigenin in hyperuricemia and renal injury, the study is interesting, well designed and well written, but the following points should be addressed:

1.      In figure 1 D-I, if we compare UA levels in NC groups between Day0 to day 2 and 6, we will notice significant decrease in UA levels in Urine, and the same applies to creatinine levels in urine. Is there any justifications for this observation? And it should be discussed in the manuscript. This point is raised because if we compare the UA levels and Cre levels with the NC in day 0 the whole data will be affected.

  1. In figure 3 , the authors showed that Apigenin has no effect on liver function enzymes and histology at all dosage used including the APH group (100 mg/kg), this finding is not in agreement with previous study  showing that Apigenin at 100 mg/kg concentration has a drastic impact on liver enzymes and histology ( DOI: 10.1371/journal.pone.0031964 ), any justifications?, and this should be included in the discussion as well.
  2. On what basis did the researchers choose 2 and 6 days as experimental set up? In previous studies the mice used to be treated for up to 21 days (https://doi.org/10.1016/j.phymed.2021.153585)? This point needs clarification and should be included in the discussion.

4.      The Apigenin has been extensively studied before for its potential hypourecemic effect, so the authors should clearly point out their scientific added value of their study in the introduction?

Reviewer 2 Report

Apigenin is a natural compound with multiple functions and low toxicity. In the present submission, the authors reported that apigenin may inhibit the development of acute hyperuricemia and its associated kidney injury through regulation of JAK2 and STAT3 pathway and UA metabolism. The findings are interesting. I have a few concerns about the current submission.

1) In Fig.2, the authors claimed that apigenin may alleviate renal injury. However, it is suggested to evaluate urinal protein levels and its relative ratio to CRE. In addition, a masson staining may provide strong evidence to illustrate the kidney injury.

2) In line 173-174, the authors claimed that apigenin molecule was relatively distant from the active center of XOD protein. Did it mean that apigenin is difficult to bind with XOD?

3) In Fig.8, please explain the meanings of ig,ip in the annotation.